# “We Don’t Have to Do Things the Way They’ve Been Done Before”; Mixed-Method Evaluation of a National Grant Program Tackling Physical Inactivity through Sport

**DOI:** 10.3390/ijerph19137931

**Published:** 2022-06-28

**Authors:** Catriona L. Rose, Katherine B. Owen, Bridget C. Foley, Lindsey J. Reece

**Affiliations:** SPRINTER Group, Prevention Research Collaboration, Charles Perkins Centre, Sydney School of Public Health, Faculty of Medicine and Health, The University of Sydney, Camperdown, NSW 2006, Australia; katherine.owen@sydney.edu.au (K.B.O.); bridget.foley@sydney.edu.au (B.C.F.); lindsey.reece@sydney.edu.au (L.J.R.)

**Keywords:** physical activity, sporting program, physical inactivity, organised physical activity, health-enhancing physical activity promotion

## Abstract

National strategies are needed to continue to promote the broader benefits of participating in sport and organised physical activity to reduce physical inactivity and related disease burden. This paper employs the RE-AIM framework to evaluate the impact of the federally funded $150 million Move it AUS program in engaging inactive people in sport and physical activity through the Participation (all ages) and Better Ageing (over 65 years) funding streams. A pragmatic, mixed-methods evaluation was conducted to understand the impact of the grant on both the participants, and the funded organisations. This included participant surveys, case studies, and qualitative interviews with funded program leaders. A total of 75% of participants in the Participation stream, and 65% in the Better Ageing stream, were classified as inactive. The largest changes in overall physical activity behaviour were seen among socioeconomically disadvantaged participants and culturally and linguistically diverse participants. Seven key insights were gained from the qualitative interviews: Clarity of who, Partnerships, Communication, Program delivery, Environmental impacts, Governance, and that Physical inactivity must be a priority. The Move It AUS program successfully engaged physically inactive participants. Additional work is needed to better engage inactive people that identify as culturally and linguistically diverse, Aboriginal and/or Torres Strait Islander and those that live in disadvantaged communities in sport and physical activities. Tangible actions from the seven key insights should be adopted into workforce capability planning for the sport sector to effectively engage physically inactive communities.

## 1. Introduction

Physical inactivity is a major public health and economic concern to global communities [1]. Despite the benefits of physical activity (PA) on population health outcomes (physical, mental, and social), improved community connectedness, and contribution to economic growth [2,3,4], limited evidence exists on population-level strategies to increase PA, particularly in communities most likely to be inactive [5]. In the 2018/19 Federal Government Budget, Sport Australia committed more than $150m to ‘Drive national sports participation and PA initiatives to get more Australian’s moving more often’ through the launch of the Australian roadmap ‘Sport 2030’, and investment through the Move It AUS grant program [6]. Support for this national plan is evident, with funding in sport identified as one of the eight best investments to tackle the growing inactivity crisis, and sport as a tool to enable active communities has been recognised and endorsed in the Global Action Plan on Physical Activity (GAPPA) [4,7]. More recently, the World Health Organization released the Fair Play advocacy brief, which signposted the necessity for PA to be a priority for all involved stakeholders to reduce the equity gap in physical inactivity [2]. Improved and equitable access to appropriate PA and sport activities must become a priority across society to address growing health inequities, made particularly evident during the COVID-19 pandemic [2,8].

Effectively evaluating the impact of population-level physical activity programs is complex and challenging. A lack of integrated evaluations and weak intervention designs are common, and few evaluations are transparent in their protocol reporting and many fail to assess program reach, omitting key process information required to make a judgement of value and translation [9,10]. The aims of this paper are to present the process and outcomes of a national government-funded national sport grant program ‘Move it AUS’ for physically inactive Australians using the RE-AIM framework. The paper will report the outcomes of the grant program’s Reach, Effectiveness, Adoption, Implementation, and Maintenance of the different components of the program in the real-world contexts of the program’s delivery [11]. This evaluation of the Move It AUS funded program will contribute to the evidence base on what works and what does not for reducing physical inactivity within communities and provide key insights on enhancing capability within the sport and recreation sectors to suggest appropriate and inclusive opportunities to recruit new target groups to participate. 

## 2. Materials and Methods

### 2.1. Funding Overview

In 2019, Sport Australia announced a $56 million investment into two funding streams, the Participation (all ages) and Better Ageing (BA, over 65 years) streams. Funding focused on engaging inactive target groups in organised sport and PA, including people living with a disability, Aboriginal and Torres Strait Islanders, women and girls, disadvantaged communities, individuals with (or at risk of) long-term conditions, and culturally and linguistically diverse (CALD) and older people.

### 2.2. Evaluation Design

The ‘Theory of Change’ guided the evaluation methods to inform how the program impacted both the capacity of the sport and PA sector (funded organisations) [12]. A logic model was created for both Participation and BA streams to ensure data collected could appropriately explain whether the program achieved these outcomes (Table 1). The use of realistic evaluation methods using the RE-AIM framework in this study aimed to understand the reasons for a certain outcome and to provide practice-relevant evidence [13]. 

A quasi-experimental mixed method design used a pre-post survey, alongside qualitative data collection with funded program operational leaders. Participant characteristics and program designs for the two streams (Participation and BA) differ and have been analysed and reported separately.

### 2.3. Ethics

The University of Sydney ethics committee granted ethics approval for this evaluation (2019/533 and 2020/250). Where required, written informed consent was attained prior to data collection. 

### 2.4. Data Collection

#### 2.4.1. Pre-Post Survey (Participant Outcomes)

Surveys were expected to be distributed by the funded organisations to all registered participants before and after their participation in the funded program, or at 6 months post initial registration. Socio-demographic data were collected from participants including postcode, which was used to classify both socioeconomic status using the Socio-Economic Indexes for Areas (SEIFA) [14], and remoteness using the Accessibility and Remoteness Index of Australia (ARIA+) [15]. This survey data were used to inform on the reach, effectiveness, and maintenance aspects of programs in achieving the aims of the funding.

The primary outcome of the survey data was meeting PA guidelines status, assessed using the validated single item measures for children 5–17 [16] and adults 18+ years old [17]. The definition for physically inactive were adults who were not completing 30 mins of PA on 5 or more days per week, and for children 60 mins of PA on 7 days per week [4]. Secondary outcomes, including organised sport participation, were aligned where possible with existing validated or accepted measures [18]. 

#### 2.4.2. Qualitative Interviews (Organisational Outcomes)

All Move It AUS grant funded organisations were invited to participate in a 30–45 min qualitative interview (Appendix A). A purposive convenience sample of organisations was recruited, and a nominated leader from each organisation participated in the interview at the conclusion of the program delivery. Interview questions related to the RE-AIM framework and provided perspectives on the process and outcomes of program delivery (Appendix A). The interviews were conducted and recorded online using Zoom (ZoomVideo Communications Inc., San Jose, CA, USA, 2016). 

### 2.5. Data Analysis

Participants’ demographic characteristics were calculated using descriptive statistics, including frequencies and proportions. Logistic regression models were used to determine whether the pre/post timepoint was associated with meeting physical activity guidelines and sport participation (at least twice per week). Model 1 is unadjusted and model 2 adjusts for age, sex, language, remoteness, and socioeconomic status. All analyses were performed in SAS Enterprise Guide 9.4 (SAS Institute, Cary, NC, USA). 

All audio recordings of the qualitative interviews were transcribed verbatim for analysis by a professional transcription company (Way With Words). Framework Analysis was deemed an appropriate approach to analyse qualitative data due to the systematic nature of the semi-structured interviews [19]. Interview transcriptions were analysed using the Framework Analysis approach in NVivo software (NVivo 12 Plus). Once familiarised with the transcripts, the research team conducted an iterative process that identified codes and sub-codes within the interviews to form a thematic scheme of the data (Figure 1). The RE-AIM framework was applied to the coded themes to understand the areas in which the grant programs were successful, or could be improved, in achieving the aims of the funding.

## 3. Results

In total, 88 diverse organisations were funded to deliver activities from July 2019 to July 2020 (Participation stream, *n* = 62), or July 2019 to July 2020 (BA stream, *n* = 26). Due to the impact of the COVID-19 pandemic, 32 of the funded programs were provided an extension beyond the planned completion date in June 2020. The 62 organisations in the Participation stream engaged 495,528 people, with 43,638 participants participating across 26 BA funded programs. A small proportion of participants responded to the pre/post survey, with 3483 (0.8%) and 6687 (15.3%) of Participation and BA stream participants responding to the survey, respectively. The results for each funding stream are presented separately herein, using the RE-AIM framework. 

### 3.1. Reach

Funded programs successfully reached target groups (Table 2) and inactive populations. A total of 75% of participants in Participation programs and 77% of participants in BA programs did not meet PA guidelines at baseline. However, there was low representation in some key target groups within the data set of inactive participants, including Aboriginal and or Torres Strait Islanders (5% Participation, <1% Better Ageing), CALD participants (11% Participation, 11% BA), and those living in outer regional/remote communities (7.6% Participation, 12% BA).

### 3.2. Effectiveness

43% of participants in the Participation stream reported increases in PA behaviours. Participants were 19% (non-significant) more likely to meet guidelines at follow-up, compared to baseline (OR:1.19, 95% CI 0.93, 1.53) (Table 3). Weekly minutes of PA increased in the Participation stream with an increase from 447.5 min per week of organised sport and PA, to 534.7 mins per week.

There was a decline in the number of participants achieving PA guidelines in the BA programs, with participants 35% less likely to meet guidelines at follow-up, compared to baseline (OR: 0.65, 95% CI 55, 0.76) (Table 3). This was particularly evident for those in the lowest SEIFA categories and those speaking a language other than English at home (Table 3). However, once engaged in the program, older adults in the BA program that spoke a language other than English at home typically spent more time in the funded activity (115 min) than native English speakers (100 min). Of all the older adults engaged, 27% also reported significant improvements in their balance after participation in the funded program.

The qualitative data also evidenced that targeted approaches to engage and deliver appropriate activities to new target groups was in engagement. Appropriate program design was also necessary to effectively retain participants to achieve PA guidelines through continued participation.

### 3.3. Adoption

A diverse range of 88 organisations were funded through the Move it AUS grant program, including 35 national sporting organisations (29 Participation, 6 BA), 7 state sporting organisations (4 Participation, 3 BA), 30 non-government organisations (22 Participation, 7 BA), 4 educational organisations (all Participation), 4 clinical organisations (all BA), and 8 local city councils (2 Participation, 6 BA). 

The organisations were at different levels of readiness, which impacted the adoption and integration of the program within the organisations’ strategies for long term results. Our findings showed that organisations with existing internal buy-in from leaders were more likely be using the support from Sport Australia to scale-up an idea already in place, rather than scoping out a pilot to test the feasibility of a new product (Appendix B). The importance of a strong organisational commitment and the integration of positive internal communication supporting the funded activity was reported as critical to the successful adoption within organisations. 

### 3.4. Implementation

The qualitative analysis of the interviews with program leads was synthesised into seven key insights (Appendix B):Clarity of who organisations aimed to reach was provided in the Move It AUS grant guidelines informed program design, recruitment, and delivery to overcome barriers specific to the nominated target groups.Partnerships were recognised as a mode of working synergistically to reach new audiences or provide new offerings designed to reducing physical inactivity through new target groups.Communication was redefined externally to emphasise the fun, social, and non-competitive aspects of sport participation and internally to advocate for internal buy-in for the funded activity and new target audience.Program designs included a traditional or modified sport, the provision of educational or capacity building resources, or a multifaceted approach. High quality program deliverers and program flexibility were central to the effective implementation and adoption of funded programs.COVID-19 disruptions forced funded organisations to pivot online, which impacted reach and program delivery both positively and negatively. Although this time enabled organisations to reflect on improving key aspects of delivery, it also emphasised the importance social connections in project delivery.Governance from Sport Australia allowed organisations to try new approaches in recruiting target groups and legitimised internal commitment to these new strategies.Participation strategies to reduce physical inactivity were recognised as a priority across the sport ecosystem despite competing priorities for resourcing within funded organisations.

### 3.5. Maintenance

There was a 5% increase in participants in the Participation stream that was seen in those that “don’t know” whether they will drop out of sport or PA after the program, with a reduction in the proportion of participants who had already dropped out by 8.7%. A total of 91% of participants in the BA stream reported that they were planning to continue their current sports and physical activities at the post time point. This suggests there may be an impact of retention in programs, despite the impact of COVID-19 on participation opportunities.

The sustainability of programs was a common concern within funded organisations (coded under “program delivery” and “governance”) (Figure 1). Most interviewees reported that an extended grant delivery time would be preferable to allow consideration for mechanisms for sustainability. Other solutions presented by organisations included effective partnerships and continued collection of data on the impact of the program to support future grant applications and strategic directions (Figure 1).

## 4. Discussion

This study aimed to evaluate how a national grant program reached and engaged inactive communities in sport and physical activity, and to understand the impact on the capability and capacity of the sport sector to meet the needs of inactive communities. This is the first time that a nationally funded grant program in Australia has specifically targeted inactive participants and results suggest that the clear focus and strategic direction of the grant program was successful in recruiting inactive people. Despite the impact of COVID-19, the programs successfully demonstrated how organised sport can reach inactive populations and investment in sport can achieve health outcomes for these populations [5]. Seven key insights were synthesised from the results, providing an improved understanding on what works and what does not when designing and implementing PA and sport initiatives for inactive populations. Investments in “sport and recreation for all” have been listed as one of the Eight Best Investments to reduce physical inactivity [3,7]. This evaluation will inform policies that may better support sporting organisations as health promotion, supporting on this directive. 

A major barrier to reducing physical inactivity is the initial engagement and reach to recruit key inactive groups [5,20]. The targeted approach of the Move It AUS grants were based on findings reported by both GAPPA, and through Sport Australia’s AusPlay data, which identified key inactive population groups [4,6]. Clarity enabled organisations to create a strategic focus and unified approach that guided all aspects of program delivery, helped identify key partners, and informed communication strategies. Funded organisations tried new communication strategies to reach new target groups that were like those found in other studies, including word of mouth, local marketing, and cross-promotion through partnerships [5,21]. Communication of sport as accessible to all, not just those that have experience in sporting activities is important and challenges perceived barriers to participation. However, methods to better engage primary target groups (including CALD, rural communities, and indigenous Australians) into organised PA and sport programs are still required [2]. 

Program design should begin with clearly defined target groups and include co-design where possible [5]. Challenging pre-conceived notions of sport as intimidating or out-of-reach through the development of beginner-friendly, non-competitive, and social options was reported as critical in engaging inactive participants. Core components of delivery should also include flexibility in delivery, embedded social opportunities, and skilled and qualified staff. The impact of a champion of the program within the organisation was emphasised by Ooms (et al., 2015) and signposts the importance of identifying and training skilled volunteers to deliver programs to ensure participant engagement and enjoyment [5]. Involvement in the funded PA programs presented an opportunity for community members to connect and create support networks. Sport and organised PA also have added benefits of group delivery, which have a positive impact on social and mental health across the lifespan [22]. Using sport as both a social connector and vehicle to achieve PA targets will be particularly relevant in recovering from the detrimental social, physical, and mental effects of experiencing various lockdowns and periods of social distancing due to COVID-19 [23,24].

There was a reduction in the proportion of participants achieving PA guidelines in the BA stream, most significantly observed in the most disadvantaged and CALD participants. Research has found that the effect of COVID-19 on population participation in PA was not equal, and further work is required to address the widening equity gap in PA, particularly in the return to sport and organised PA after COVID-19 [8,25]. However, our findings also reported that once engaged, CALD communities participating in BA programs engaged for an average of 15 mins longer each week in funded activities than English speakers. This evidences the feasibility of tailored programs as a gateway to achieve physical activity guidelines for inactive minority groups. Barriers to participation are greater for these minority groups, therefore, socioecological models for understanding participation rates may be used to better understand how to design and deliver effectively tailored organised sport and PA [26,27]. 

Recently released by World Health Organisation, the Fair Play advocacy brief has called for greater cross-sectoral collaboration to better engage and retain inactive target groups [2]. Our evaluation found that despite the complicated process of aligning strategic objectives organisations, partnerships were a critical factor to the success in reaching and delivering sport and PA programs to new target groups. Casey et al. (2011) made the case for long-term commitments in funding strategies and partnerships to provide sustainability, which was echoed from program providers concerned about resources required to maintain delivery [28]. Similarly, Staley et al. (2019) found that addressing inactivity through sport requires collaboration and support across multiple levels of the ecosystem [20]. Cross-sectoral collaboration is instrumental in reaching specific inactive target groups and should be embedded in future initiatives to support sustained delivery and fair access to sport and PA opportunities across the lifespan [2,5,28,29]. 

### Limitations

Funded organisations were responsible for disseminating the surveys. Some surveys were not able to be determined at either time point but are included in the aggregate total of respondents. Some organisations modified the data collection methods during program delivery due to unforeseen practicality implications such as language barriers and available resourcing. 

Findings would be strengthened in future with information on maintenance and the long-term impact of the Move It AUS grants, particularly in the absence of COVID-19 implications. Participation in the qualitative surveys was voluntary, and the possibility for self-selection bias should be noted.

## 5. Conclusions

The strong engagement of inactive people in Move It AUS funded programs demonstrates the success and acceptability of targeted interventions in engaging inactive people to reduce physical inactivity and improve health for all. The sport sector is motivated and mobilised to be part of the solution to physical inactivity, and integration of the seven key insights from this study can inform future policies and opportunities supporting sporting programs for inactive populations in the future. Utilising grant programs to broaden the population engagement throughout sport and enhance the capability of the sport sector is one strategy for increasing population levels of PA and understanding the unique contribution sport makes to our local communities.

## Figures and Tables

**Figure 1 ijerph-19-07931-f001:**
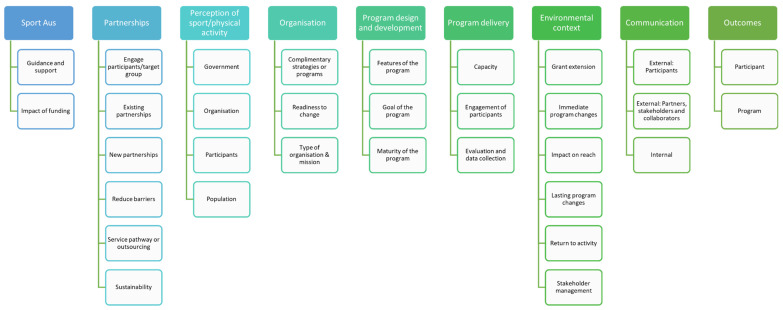
A thematic scheme of the codes and sub-codes identified within the qualitative interview data.

**Table 1 ijerph-19-07931-t001:** Sport Australia Move It AUS Participation and Better Ageing (BA) Logic Model.

Inputs	Activities	Outputs	Outcomes		
			Short (June 2019–June 2021)	Medium (July 2021–June 2023)	Long-Term (July 2023–)
**$28 m (Participation) and $22.9 m (BA) Federal investment over 4 years (Participation** **Federal investment for marketing support** **2 FTE Sport Australia staff members plus in-kind cross agency support** **Evaluation support (University of Sydney)** **Sport 2030** **Sport Australia strategic plan** **Move it AUS campaign**	62 (Participation) and 26 (BA) successful projects awarded Marketing toolkit developed for project leads & associated partners Monitoring & performance toolkit developed for project leads Case study toolkit designed 3 sector workshops developed with funded projects Move it AUS program evaluation developed by SPRINTER (Sydney Uni)	**Sport and Physical Activity Sector**			
		62 Participation and 26 BA projects funded across sport & physical activity sector 8 (Participation), and 10 (BA) marketing case studies developed Recognition of indirect beneficiaries engaged Move It AUS grants delivered in regional and remote areas Target populations engaged through Move it AUS grants Increased understanding of the sport and physical activity delivery partner network Increase capability of sport and physical activity partner partners Enhance the partnerships of sport and physical activity partners Independent National Evaluation report (University of Sydney)	Engage new populations (inactive and active, all ages) Gain in-depth insights into participation behaviours across active, inactive, and target populations Understand reasons for drop-out & barriers to physical activity Increased capacity & understanding from sport & PA sector to tackle physical inactivity through piloting of innovative projects Improved collaboration between government departments, Sport AUS and delivery partners Contribution to the evidence base on physical inactivity.	Partners commit to reducing % inactive people by 2030 Reduce the proportion of people who drop out/lost to follow up in physical activity opportunities Ongoing & continual reflection by sport and PA sector to focus on understanding and meeting needs of inactive populations (governance & quality control) Establish new and cement existing cross-agency partnerships Increased variety and availability of physical activity opportunities for local communities Continued contribution to building and listening to the evidence base across the sector for what works (and what does not work) in reducing physical inactivity in people	Partners commit to reducing % inactive people by 2030 Increase number of partners who commit to narrowing the equity gap in population participation Increase sector capability to deliver inclusive, robust and sustainable physical activity opportunities Continued contribution to building and listening to the evidence base across the sector for what works (and what does not work) in reducing physical inactivity in people
		**Participants**			
		Inactive people engaged across 62 (Participation) and 26 (BA) Move it AUS grant projects Increased awareness of physical activity guidelines among participants Increased awareness of the Move It AUS campaign among participants	Increased self-efficacy of individuals to increase and maintain physical activity behaviours Enhanced recognition and awareness of funded projects by target population Positive attitudes towards physical activity and sporting opportunities Increased awareness among people over 65 of the physical activity guidelines and benefits of physical activity in the prevention and management of chronic disease. Increased awareness among people over 65 of the importance of physical activity in improving physical strength and balance to reduce the risk of falls.	Increased self-efficacy of individuals to increase and maintain physical activity behaviors Initiation and maintenance of ‘new’ physical activity behaviour Initiation and maintenance of physical activity levels by active people Improved quality of life of people over 65 engaged in physical activity through enhanced physical, emotional, and social wellbeing.	Increased proportion of people meeting PA guideline Contribute to population reduction of physical inactivity Equity gap in population participation reduced

**Table 2 ijerph-19-07931-t002:** Demographic characteristics of participants in the Participation and Better Ageing stream across timepoints.

	Participation Stream						Better Ageing Stream					
	Pre		Post		All		Pre		Post		All	
	N	%	N	%	N	%	N	%	N	%	N	%
All persons	1410	100	1328	100	3837	100	3351	100	2649	100	6687	100
Age category												
0–17	536	43.4	730	58.2	1604	45.1	-	-	-	-	-	-
18–34	233	18.9	141	11.2	526	14.8	-	-	-	-	-	-
35–44	230	18.6	190	15.2	795	22.4	-	-	-	-	-	-
45–54	143	11.6	138	11.0	451	12.7	-	-	-	-	-	-
55–64	76	6.2	46	3.7	143	4.0	63	20.0	468	27.7	594	25.2
65+	17	1.4	9	0.7	34	1.0	252	80.0	1220	72.3	1762	74.8
Sex												
Male	543	38.8	592	44.8	1347	35.5	1001	32.5	382	21.7	1433	27.5
Female	824	58.8	705	53.3	2383	62.8	2079	67.5	1382	78.3	3771	72.5
Prefer not to say	34	2.4	25	1.9	64	1.7						
Indigenous												
Yes, Aboriginal	81	5.9	64	4.8	179	4.8	57	1.8	6	0.3	63	1.2
and/or Torres Strait												
Islander												
No	1273	92.2	1230	93.2	3492	93.4	3104	98.2	1821	99.7	5315	98.8
Prefer not to say	26	1.9	26	2.0	66	1.8						
Primary language												
English	1229	87.7	1121	85.2	3337	88.9	3086	97.0	1610	72.8	5046	87.3
Other	173	12.3	194	14.8	417	11.1	94	3.0	601	27.2	736	12.7
Employment												
Employed	275	28.7	310	30.6	1177	43.1	1085	35.0	434	16.8	1563	25.9
Unemployed	88	9.2	62	6.1	172	6.3	119	3.8	802	31.1	931	15.4
Student	359	37.5	529	52.2	943	34.5	11	0.4	2	0.1	13	0.2
Pension/welfare	186	19.4	85	8.4	291	10.7	328	10.6	280	10.9	664	11.0
Retired	24	2.5	4	0.4	36	1.3	1446	46.6	997	38.7	2689	44.5
Other	26	2.7	24	2.4	113	4.1	113	3.6	64	2.5	183	3.0
Location												
Major Cities	753	58.2	802	69.4	2314	65.9	1363	44.4	1720	81.1	3357	60.5
Inner Regional	392	30.3	280	24.2	908	25.9	1071	34.9	282	13.3	1404	25.3
Outer Regional and remote	149	11.5	73	6.3	290	8.3	637	20.7	118	5.6	788	14.2
Socioeconomic status												
1st	338	26.2	371	32.2	813	23.2	642	20.9	298	14.0	987	17.8
2nd	222	17.2	187	16.2	655	18.7	920	29.9	459	21.6	1461	26.3
3rd	395	30.6	276	24.0	974	27.8	664	21.6	623	29.4	1366	24.6
4th	337	26.1	317	27.5	1061	30.3	847	27.6	741	34.9	1740	31.3
Health condition												
Yes	488	36.2	320	25.3	810	31.0	1451	50.6	1107	44.5	2564	47.8
No	859	63.8	943	74.7	1807	69.0	1416	49.4	1381	55.5	2797	52.2

Note: “All” column includes those who could not be classified as pre or post.

**Table 3 ijerph-19-07931-t003:** Odds of meeting physical activity guidelines across timepoints in the Participation and Better ageing funding stream.

	Participation				Better Ageing			
	Unadjusted Proportions Meeting Physical Activity Guidelines		Unadjusted Odds Ratio for Meeting Physical Activity Guidelines	Adjusted Odds Ratio for Meeting Physical Activity Guidelines	Unadjusted Proportions Meeting Physical Activity Guidelines		Unadjusted Odds Ratio for Meeting Physical Activity Guidelines	Adjusted Odds Ratio for Meeting Physical Activity Guidelines
	Pre (%)	Post (%)	OR (95% CIs)	OR (95% CIs)	Pre (%)	Post (%)	OR (95% CIs)	OR (95% CIs)
All persons	25.0	27.7	1.15 (0.95, 1.39)	1.19 (0.93, 1.53)	35.1	21.0	0.49 (0.43, 0.57)	0.65 (0.55, 0.76)
Age category								
0–17	13.3	13.6	1.03 (0.69, 1.54)	1.21 (0.65, 2.22)				
18–34	30.5	40.4	1.55 (1.00, 2.40)	1.72 (0.95, 3.11)				
35–44	40.9	40.0	0.96 (0.65, 1.43)	0.93 (0.6, 1.43)				
45–54	32.9	48.6	1.93 (1.19, 3.12)	1.81 (1.05, 3.12)				
55–64	35.5	34.8	0.97 (0.45, 2.08)	1.45 (0.5, 4.23)				
65+	17.7	66.7	9.33 (1.45, 60.21)					
Sex								
Male	21.8	24.3	1.15 (0.85, 1.56)	1.24 (0.84, 1.83)	39.2	20.4	0.40 (0.30, 0.53)	0.60 (0.44, 0.82)
Female	27.6	31.5	1.21 (0.95, 1.54)	1.32 (0.95, 1.83)	33.1	21.2	0.54 (0.46, 0.64)	0.65 (0.54, 0.79)
Indigenous								
Yes, Aboriginal	11.1	41.5	5.67 (1.98, 16.22)	40.77 (3.75, 443.83)	28.1	16.7	0.51 (0.06, 4.73)	2.4 (0.05, 114.79)
and/or Torres								
Strait Islander								
No	26.7	27.5	1.04 (0.86, 1.26)	1.08 (0.84, 1.39)	34.9	21.5	0.51 (0.45, 0.58)	0.65 (0.55, 0.76)
Primary language								
English	25.8	29.9	1.23 (1.00, 1.5)	1.29 (1, 1.67)	35.1	33.7	0.94 (0.83, 1.07)	0.71 (0.60, 0.84)
Other	20.4	16.7	0.78 (0.44, 1.38)	0.94 (0.42, 2.10)	26.9	7.8	0.23 (0.13, 0.40)	0.18 (0.10, 0.34)
Employment								
Employed	41.4	39.9	0.94 (0.67, 1.31)	1.07 (0.74, 1.55)	34.1	35.2	1.05 (0.81, 1.36)	0.76 (0.53, 1.08)
Unemployed	26.4	17.0	0.57 (0.25, 1.3)	0.39 (0.12, 1.22)	40.2	13.0	0.22 (0.15, 0.34)	0.39 (0.23, 0.65)
Student	10.7	10.8	1.02 (0.63, 1.63)	1.26 (0.71, 2.21)	36.4	50.0	1.75 (0.08, 36.29)	
Pension/welfare	15.1	38.8	3.58 (1.98, 6.48)	3.29 (1.75, 6.2)	27.3	34.6	1.41 (1.00, 2.00)	0.93 (0.51, 1.71)
Retired	41.7	50.0	1.4 (0.17, 11.68)		36.3	36.7	1.01 (0.86, 1.2)	0.80 (0.63, 1.02)
Other	46.2	52.2	1.27 (0.41, 3.92)		38.6	28.6	0.64 (0.21, 1.89)	0.25 (0.01, 6.86)
Location								
Major Cities	28.3	28.0	0.99 (0.77, 1.27)	0.8 (0.56, 1.13)	35.7	23.8	0.56 (0.48, 0.66)	0.58 (0.48, 0.7)
Inner Regional	23.1	35.4	1.83 (1.28, 2.62)	2.24 (1.46, 3.43)	35.3	35.2	1.00 (0.76, 1.31)	0.66 (0.42, 1.03)
Outer Regional	31.5	30.4	0.95 (0.51, 1.78)	0.92 (0.46, 1.84)	33.7	47.0	1.75 (1.17, 2.61)	1.56 (0.94, 2.59)
and remote								
Socioeconomic status								
1st	21.5	21.8	1.02 (0.69, 1.50)	1.16 (0.72, 1.88)	36.3	23.7	0.54 (0.40, 0.74)	0.31 (0.17, 0.56)
2nd	24.5	38.7	1.95 (1.23, 3.08)	1.96 (1.10, 3.50)	34.1	22.5	0.56 (0.43, 0.73)	0.68 (0.47, 0.98)
3rd	26.4	32.4	1.34 (0.93, 1.94)	1.01 (0.59, 1.70)	32.6	29.0	0.84 (0.66, 1.07)	0.90 (0.67, 1.22)
4th	35.9	33.6	0.91 (0.63, 1.31)	1.06 (0.63, 1.78)	37.7	28.4	0.66 (0.53, 0.81)	0.59 (0.45, 0.76)
Health condition								
Yes	21.5	28.4	1.45 (1.03, 2.03)	1.56 (1.03, 2.35)	30.3	26.5	0.83 (0.70, 0.99)	0.69 (0.53, 0.89)
No	27.2	26.9	0.98 (0.78, 1.24)	1.03 (0.74, 1.43)	39.6	29.0	0.63 (0.53, 0.73)	0.62 (0.50, 0.77)

## Data Availability

Not applicable.

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
