# Peer review of "“We Don’t Have to Do Things the Way They’ve Been Done Before”; Mixed-Method Evaluation of a National Grant Program Tackling Physical Inactivity through Sport"

_ijerph, 2022, doi:10.3390/ijerph19137931_

Round 1
Reviewer 1 Report
Congratulations on your work. Given the quality, scope and depth of your work it is very difficult to find significant suggestions, however, it could be suggested to include data referring to the long-term maintenance of BP and AB and not to refer the data exclusively to the time when the BP programme ended.
It would also be interesting to establish correlations between all the variables described and the long-term maintenance of BP and AB habits.
It may also raise doubts, as you seem to indicate in the limitations section, that the surveys were "controlled" by the funded organisations, which may do more than what you indicate that they can influence the results of the surveys to justify their activities. But of course, this is only an assumption that we hope has not been made.
In any case I congratulate the authors and wish them all the best for the future.
Author Response
Response 1: We appreciate the reviewers feedback but only have data to include that was collected during program delivery, and therefore long-term maintenance impact may not be inferred. Furthermore, we are unsure what the reviewer is referring to regarding the acronyms BP and AB.
Response 2: As mentioned in response 1, we are unsure what is meant by BP and AB habits.
Response 3: We appreciate this comment, however the survey was created by the research team, and hosted on an online survey platform controlled by the research team. As such, there was limited involvement of organisations in the delivery of the survey to participants. However, where the survey was modified, the research team worked with organisations to ensure the validity of the surveys was upheld in the modified version.
Reviewer 2 Report
Thank you fo letting me review the manuscript about how a national grant program reached and engaged innactive communities in physical activities.
In general, I think is a well-design and well-written study. I understand that takes a long time to stay in touch with all the organizations and to get all the surveys in order to include most representative data.
After a carefully evaluation I only have 2 suggestions about the manuscript.
- Pleasre revise the acronyms through the entire manuscript.
My main concern, despite of the fact that the study is about foundations and physical activity, is that this kind of study has not a high interest soundness and is not really interesting for readers due to is a very specific topic in a specific country and population.
But, again, I think that the authros did a very good job.
Author Response
Response 1: We have gone through the manuscript to clarify the use of acronyms and remove them where unnecessary. Page 1, line 43-page 2, line 44 – included clarity on the funding program (Move It AUS), within the context of broader government funding. Page 2, Line 46 – included a description of GAPPA. Page 2, line 70 – included acronym for Better Ageing (BA).
Response 2: The authors accept this comment, however we feel that these findings are of great practical applicability to many sporting and physical activity organisations looking to engage new, inactive cohorts. It is hoped that learnings from this type of research, even if conducted in high-income countries, may be applied in the future to other settings including low- and middle-income countries, to promote sport and physical activity participation.
Reviewer 3 Report
Evaluation of the federally funded $150 million Move it AUS program in engaging inactive people in sport and physical activity through the Participation (all ages) and Better Ageing (over 65 years) funding streams. The study evaluated how a national grant program in Australia reached and engaged inactive communities in sport and physical activity, and to understand the impact on the capability and capacity of the sport sector to meet the needs of inactive communities. They are giving new strategies through evaluation of seven key points how to improve further funded programs, their implementation, processing, and maintenance of the individuals in the programme for this country. They have evaluated the program in 88 diverse organisations that were funded to deliver activities from 2019 to 2020 engaging 495,528 people, with 43,638 participants participating across funded programs. Response: Evaluation like that of the Programe like that in specific country was not tested until nowThe conclusions consistent with the evidence and arguments
presented and do they address the main question posed.
The references are appropriate.
Author Response
Response 1: The authors appreciate this feedback
Reviewer 4 Report
Thank you for the opportunity to review this valuable work.
The theme of this study is interesting and important. The national strategies or policies are increasingly valuable options for preventing inactivities in the latest decade. This well structured study was questioned the benefit of national promotions and their effectiveness, and demonstrated important findings on this field.
I don't have any comments for major revisions. Congratulations to the authors for their effort
Author Response

(The authors gave the same response as above.)

Round 2
Reviewer 2 Report
The authors considered all my suggestions. No additional comments.